# UAV-Borne Mapping Algorithms for Low-Altitude and High-Speed Drone Applications

**DOI:** 10.3390/s24072204

**Published:** 2024-03-29

**Authors:** Jincheng Zhang, Artur Wolek, Andrew R. Willis

**Affiliations:** 1Department of Electrical and Computer Engineering, University of North Carolina at Charlotte, Charlotte, NC 28223, USA; arwillis@charlotte.edu; 2Department of Mechanical Engineering and Engineering Science, University of North Carolina at Charlotte, Charlotte, NC 28223, USA; awolek@charlotte.edu

**Keywords:** UAV, drone, mapping, SfM, stereo reconstruction, 3D reconstruction

## Abstract

This article presents an analysis of current state-of-the-art sensors and how these sensors work with several mapping algorithms for UAV (Unmanned Aerial Vehicle) applications, focusing on low-altitude and high-speed scenarios. A new experimental construct is created using highly realistic environments made possible by integrating the AirSim simulator with Google 3D maps models using the Cesium Tiles plugin. Experiments are conducted in this high-realism simulated environment to evaluate the performance of three distinct mapping algorithms: (1) Direct Sparse Odometry (DSO), (2) Stereo DSO (SDSO), and (3) DSO Lite (DSOL). Experimental results evaluate algorithms based on their measured geometric accuracy and computational speed. The results provide valuable insights into the strengths and limitations of each algorithm. Findings quantify compromises in UAV algorithm selection, allowing researchers to find the mapping solution best suited to their application, which often requires a compromise between computational performance and the density and accuracy of geometric map estimates. Results indicate that for UAVs with restrictive computing resources, DSOL is the best option. For systems with payload capacity and modest compute resources, SDSO is the best option. If only one camera is available, DSO is the option to choose for applications that require dense mapping results.

## 1. Introduction

UAVs, also known as drones, have transcended conventional applications to become indispensable tools across an array of disciplines, from environmental monitoring and precision agriculture to disaster response and infrastructure inspection. At the heart of their efficacy lies the sophisticated interplay between UAVs and mapping algorithms, which serve as the backbone for converting raw sensor data into coherent, high-fidelity maps. These algorithms play a pivotal role in navigating complex terrains, extracting meaningful information, and ensuring precise localization of the UAV in real time. From traditional photogrammetry to advanced techniques like Simultaneous Localization and Mapping (SLAM), these algorithms continuously evolve to meet the diverse demands of UAV applications ranging from agriculture and forestry to disaster response and urban planning.

Mapping algorithms for UAVs are significantly influenced by flight altitude, dictating the scale of environmental perception and mapping capabilities. For high-altitude flights, the imagery changes between successive frames are slower than for low-altitude flights, which allows more overlap/correspondence between successive frames. However, as altitude increases, challenges such as reduced sensor performance, diminished feature visibility, and heightened geometric distortions emerge. While 3D or 2D laser scanners generate effective terrain models, their weight and sensitivity to ground proximity pose challenges. Compact depth-sensing devices, though commercially available, often fall short in operational range. Camera-based mapping systems, while lightweight and scalable, face accuracy challenges at high altitudes due to reduced texture and discernible features. This limitation hampers feature tracking and matching, impacting overall mapping algorithm performance. High-altitude flights also amplify drift and uncertainty in UAV trajectory estimation, particularly affecting SLAM algorithms relying on sensor fusion. The accumulation of errors over time compromises poses estimations, emphasizing the critical consideration of flight altitude in optimizing mapping algorithm outcomes. This article focuses on multirotor UAVs and analyzes UAV algorithm performance at altitude ranges from 12 m to 20 m from the ground, which is considered to be “low-altitude” in this article. Investigations for this context provide an analysis of key sensor options and their strengths and weaknesses. Specific recommendations are also provided for light-duty UAVs (U.S. military UAS Group 1).

Mapping algorithms tailored for high-speed UAVs address the specific demands of dynamic and rapid flight scenarios. Real-time operation in these contexts is imperative, necessitating synchronization and integration of data from diverse sensors such as LiDAR, cameras, and inertial measurement units (IMUs). Adaptive navigation is equally crucial to accommodate the UAV’s swift maneuvers and maintain mapping precision. Overcoming challenges related to large distances covered between sensor readings during high-speed flights is essential for achieving precise mapping results. Additionally, robustness in the face of environmental variability, including changes in lighting, weather conditions, and terrains, is vital. High-speed UAVs, integral in applications like surveillance and emergency response, benefit from ongoing advancements in mapping algorithms. These improvements enhance effectiveness, allowing UAVs to navigate rapidly changing environments and deliver precise and timely mapping outcomes. This article analyzes multirotor UAV algorithm performance for speed ranges from 15 m/s to 20 m/s which corresponds to the maximum speed for typical commercially available platforms in this Group [1].

In recent decades, research investigating methods for 3D reconstruction from images has thrived. Examples of approaches include Structure-from-Motion (SfM) algorithms [2,3,4,5,6,7] and stereo reconstruction (Stereo3D) algorithms [8,9,10,11,12]. These approaches are the algorithms that can be used as components of a SLAM system. SfM and Stereo3D differ in both computation methods and output formats. SfM algorithms analyze a sequence of 2D images from a camera and estimate the relative motion of the camera and the geometric structure of the observed 3D scene. Motion estimates include the camera pose, i.e., position and orientation, at each recorded image and the scene 3D structure observed in each image. Stereo3D estimates 3D scene structure from a pair of 2D images captured simultaneously by two cameras with known relative positions. Depth information is derived from the correspondence of observed scene points between the two images. While both SfM and Stereo3D target 3D scene reconstruction, they excel in different applications and scenarios.

The contributions of this article include:A comprehensive analysis outlining the strengths and limitations of state-of-the-art SfM and stereo reconstruction algorithms;A benchmark of the geometry accuracy and computation speed of various mapping algorithms;A theoretical foundation for sensor selection tailored to low-altitude and high-speed UAV mapping applications;A technical approach for extracting high fidelity geometric models from Cesium Tile data to perform analysis on 3D mapping and odometry algorithms;An innovative approach to simulate realistic flights, utilizing Unreal Engine for high-realism environment synthesis, Cesium plug-in for geographical context, AirSim for vehicle dynamics, and PX4 Autopilot for precise vehicle control.

These contributions provide researchers new insight into how to best adopt mapping technologies for their UAV design in low-altitude and high-speed drone applications.

An initial discussion evaluates the theoretical suitability of a wide variety of sensors for this application and eliminates many sensors from candidacy for various technical reasons. Subsequent evaluation of algorithms is contingent on the proposed selection of best-practice sensors for this context.

Mapping algorithm analysis surveys current state-of-the-art real-time reconstruction algorithms suited to the sensors that were previously identified as appropriate for low-altitude and high-speed multirotor UAV mapping applications. From a wide array of possible algorithms, three were evaluated: (1) Direct Sparse Odometry (DSO) [13], (2) Stereo Direct Sparse Odometry (SDSO) [14], and (3) Direct Sparse Odometry Lite (DSOL) [15]. While many algorithms are available in the literature, the selected algorithms provide a representative sampling of reconstruction methods for the recommended camera sensors.

## 2. Related Work

This article compares three methods for 3D mapping in terms of their suitability for use on Group 1 UAVs at high-speed, low-altitude flight. Discussion of current sensing options indicates that camera-based methods are well-suited to this application. Experiments use a simulated environment to evaluate leading camera-based methods. For these reasons, a review of the related literature to this article is divided into three parts:A comparison of SfM and stereo3D reconstruction methods including recent leading implementations of these methods;A compact review of three 3D-from-images algorithms;A review of different 3D simulation options for developing and evaluating these mapping algorithms for the context of low-altitude high-speed flight.

A comprehensive literature review motivates the methodology and experimental approach for this article. Specifically, the choice of mapping algorithms analyzed and the simulation environment used was based on a comprehensive review of candidate solutions.

### 2.1. Structure-from-Motion vs. Stereo Reconstruction

Structure from Motion (SfM) and stereo reconstruction are two leading techniques employed in 3D reconstruction. This subsection describes the principles of both techniques to provide insights into their distinctive attributes and how they relate to high-speed low-altitude mapping applications.

#### 2.1.1. Structure-from-Motion

SfM [16] is the process of reconstructing a 3D structure from its projections into a series of images taken from different viewpoints. It leverages the relative movement between a camera and objects in a scene to reconstruct the 3D structure. SfM estimates the camera poses and the spatial arrangement of points in the scene by analyzing the changes in perspective across multiple images. SfM has been extensively studied and applied in diverse fields, including 3D modeling [17,18], augmented reality [19,20], autonomous navigation [21,22], and remote sensing [23,24]. Researchers have explored various algorithms and optimization methods to enhance the accuracy [2,25,26,27] and efficiency [4,6,7,28] of SfM, making it a robust solution for scenarios where camera poses change dynamically, a common occurrence in high-speed low-altitude flights.

#### 2.1.2. Stereo Reconstruction

Stereo reconstruction (stereo3D) involves the process of estimating the 3D structure of a scene from a pair of 2D images captured by two cameras with known relative positions. By analyzing the disparities between the two images, stereo reconstruction algorithms can calculate the depth information of the scene points. This depth information allows for the creation of a 3D representation of the scene. Stereo3D is critical to enabling autonomous capabilities in a wide range of fields including robotics [29,30], autonomous vehicles [9,31], and 3D modeling [12,32,33].

Figure 1a illustrates the epipolar geometry of two pinhole cameras observing a 3D point M. Stereo reconstruction estimates M’s distance by analyzing its projections m and $m^{\prime }$. The baseline *B* connects camera origins CL and CR, defining epipolar geometry with epipoles e and $e^{\prime }$. The epipolar plane intersects with image planes π and $\pi^{\prime }$, forming epipolar lines. According to epipolar geometry, m in $\pi^{\prime }$ lies on epipolar line $l^{\prime }$. Depth estimation involves finding corresponding points, simplified by image rectification in Figure 1b, ensuring m and $m^{\prime }$ align. Depth *d* is estimated through triangulation represented by Equation (Equation 1), considering column differences from m and $m^{\prime }$ to the center of the left and right images, baseline *B*, focal length *f*, and pixel width δ in the rectified image sensor.
(1)$$d=\frac{Bf}{\delta \left(x-x^{\prime }\right)}$$

Figure 2 shows the theoretical dependency between the baseline parameter of a stereo camera pair and the accuracy of the depth estimates that the stereo sensor will produce. Red lines show the depth deviations associated with a ±1 pixel error in the disparity. The plot shows that the disparity decreases as a square of the depth and error increases as a square of the depth.

SfM can be computationally intensive and requires feature matching and bundle adjustment for robust results. Stereo3D, with fixed camera positions, is typically less computationally intensive and more straightforward compared to SfM. SfM systems estimate the scene structure to an unknown scale and usually require fusion with other metric data, e.g., from an IMU or a GPS sensor, to make estimated geometric measurements consistent with the real geometric scene structure. Stereo3D directly estimates the scene structure and uses the baseline distance to provide scene scale estimates that are metrically consistent with the 3D scene geometry and do not require sensor fusion to recover the unknown scale.

### 2.2. Mapping Algorithms

This article focuses on the following three representative state-of-the-art real-time algorithms to investigate their applications to multirotor UAV-borne mapping:Structure-from-Motion: Direct Sparse Odometry (DSO) [13];Stereo Reconstruction: Stereo Direct Sparse Odometry (SDSO) [14] and Direct Sparse Odometry Lite (DSOL) [15].

#### 2.2.1. DSO: Direct Sparse Odometry

DSO is a visual odometry technique that adapts SfM methods for 3D reconstruction. It directly estimates the camera motion and the sparse 3D structure of the environment from a sequence of 2D images by minimizing photometric errors. DSO differs significantly from traditional techniques by directly optimizing photometric errors in images, without relying on keypoint detectors or geometric priors. For a point, p in reference frame $I_{i}$, observed as $p^{\prime }$ in target frame $I_{j}$, the photometric error, given by Equation (Equation 2), is formulated as the weighted Sum of Squared Differences (SSD) over a small neighborhood of pixels.
(2)$$E_{pj}:=\sum_{p\in N_{p}}w_{p}{∥\left(I_{j}\left[p^{\prime }\right]-b_{j}\right)-\frac{t_{j}e^{a_{j}}}{t_{i}e^{a_{i}}}\left(I_{i}\left[p\right]-b_{i}\right)∥}_{\gamma }$$
where $N_{p}$ is the set of pixels in the SSD; $\left(t_{i},t_{j}\right)$ the exposure times of the frame $I_{i}$ and $I_{j}$; $\left(a_{i},b_{i},a_{j},b_{j}\right)$ the brightness transfer variables defined in DSO for frame $I_{i}$ and $I_{j}$, respectively, and ${∥\;\cdot \;∥}_{\gamma }$ is the Huber norm. In addition to using robust Huber penalties, a gradient-dependent weighting $w_{p}$ is applied. Further, $p^{\prime }$ stands for the projected point position of p with inverse depth $d_{p}$, given by
(3)$$p^{\prime }=\Pi_{c}\left(R\Pi_{c}^{-1}\left(p,d_{p}\right)+t\right)$$
with
(4)$$\left[\begin{array}{cc} R & t\\ 0 & 1 \end{array}\right]:=T_{j}T_{i}^{-1}$$
where $\Pi_{c}:R^{3}\to \Omega$ denotes projection, $\Pi_{c}^{-1}:\Omega \times R\to R^{3}$ denotes back-projection, c denotes the intrinsic camera parameters, and $T_{i},T_{j}\in \mathrm{SE}\left(3\right)$ are the camera poses represented by transformation matrices for frame $I_{i}$ and $I_{j}$.

To minimize the photometric error between the corresponding points in two frames, DSO incorporates a fully direct probabilistic model that jointly optimizes all model parameters, including camera motion and geometry, represented as inverse depth in a reference frame. The optimization is accomplished using the Gauss-Newton algorithm in a sliding window [35].

#### 2.2.2. SDSO: Stereo Direct Sparse Odometry

SDSO is a stereo version of DSO. In a monocular mapping system like DSO, to initialize the whole system, i.e., to track the second frame with respect to the initial one using Equation (Equation 2), the inverse depth values $d_{p}$ of the points in the first frame are required. In DSO, the points are initialized to have random depth values ranging from 0 to infinity, corresponding to a large depth variance. Unlike that, SDSO uses stereo matching to estimate a semi-dense depth map for the first frame, which significantly increases the tracking accuracy. The constraints from static stereo introduce scale information into the system. They also provide good geometric priors to temporal multi-view stereo.

#### 2.2.3. DSOL: Direct Sparse Odometry Lite

DSOL presents an enhanced version of DSO and SDSO, proposing several algorithmic and implementation improvements to significantly speed up computation. Following the same practice as DSO of defining the photometric error in Equation (Equation 2), DSOL adopts the inverse compositional alignment method [36] to perform computationally expensive calculations, i.e., the Gauss-Newton approximation to the Hessian matrix, at the pre-computation phase, which largely improves the running speed of the algorithm. Compared to DSO and Stereo DSO, key aspects of optimization in DSOL include the following: (1) utilizing an inverse compositional alignment method for frame tracking, improving accuracy and speed; (2) adapting a better stereo photometric bundle adjustment formulation compared to SDSO; (3) simplifying keyframe creation and removal criteria from DSO, allowing for better utilization of computational resources and parallel processing; and (4) implementing algorithmic enhancements to streamline the computation process, making it more suitable for real-time applications, especially in resource-constrained environments. The focus of DSOL is on mapping speed and efficiency while maintaining accuracy.

### 2.3. Aerial Simulation Solutions

There are various simulation platforms for vehicles and environments catering to the diverse needs of researchers. Gazebo [37], with its open-source nature, stands as a versatile choice, emphasizing realism and adaptability. Agilicious [38] specializes in agile quadrotor flight, providing unique applications such as drone racing. RotorS [39], integrated with the Robot Operating System (ROS), offers high-fidelity UAV simulation. Flightmare [40], part of the AirSim project, excels in simulating multiple drones for swarm robotics research. Kumar Robotics Autonomous Flight [41] addresses GPS-denied quadcopter autonomy. MIT’s FlightGoggles [42] offers an immersive experience with photorealistic graphics. AirSim, developed by Microsoft, on top of the Unreal Engine, excels in generating highly realistic perceptual simulation data in complex and dynamic environments.

An approach is proposed in [43] to reproduce real-world experiments in simulation using the AirSim open-source simulator with the Cesium Tiles plugin, allowing for large-scale 3D geometry analysis. This paper adapts the methodology and extends it with other aerial vehicle control technologies, achieving precise vehicle control in high-realism virtual models that replicate real-world contexts world.

## 3. Methodology

The overall approach for the methods of this article consists of three steps:Describe the benefits and shortcomings of various candidate sensing modalities for low-altitude high-speed mapping using Group 1 UAVs resulting in a recommendation for using one or more high-frame rate conventional camera sensors for this application (Section 3.1).Describe the simulation methods used to collect data using a highly realistic 3D environment made possible by integrating the AirSim simulator with Google’s 3D map database using the Cesium Tiles plugin for the Unreal Engine (Section 3.2).Describe the evaluation methods adopted to compare the mapping results generated from experimental flights within the simulated environment (Section 3.3).

### 3.1. Sensors for UAV Mapping

Three prominent sensor types are investigated as potential components of the UAV perceptual payload. These sensor types are listed below:LiDAR (Light Distance and Ranging) Sensors;Event Cameras;Conventional EO and IR Cameras.

Our assessment considered leading examples of each sensor that would be potentially appropriate for the high-speed low-altitude context and commercially available. The specifications of the sensors were then reviewed in terms of their ability to provide measurements that meet the requirements of UAV mapping. Based on this analysis, a determination was reached regarding the suitability of each sensor.

#### 3.1.1. LiDAR

Figure 3 shows several LiDAR sensors evaluated for inclusion in the platform payload. LiDAR sensors have emerged as a popular choice for UAV mapping applications with significant advancements in LiDAR-based techniques [44,45,46,47]. However, it was quickly determined that these devices would not be appropriate for the UAV mapping application. The shortcomings of these sensors are described in the list below:Weight: LiDAR sensors typically weigh 500 g. or more which would be equivalent to approximately 5 image sensors of 100 g.Measurement Method: LiDAR sensors measure individual 3D points at one time or a collection of 3D points using a laser line-scanning technology. In either case, a rotating mirror in the sensor scans the scene over time. Accurate integration of scan data requires motion compensation for individual 3D point measurements for mapping and geometry estimation.Measurement Speed: LiDAR sensors typically scan at low rates (10–20 Hz) which makes the capture of a complete 3D scene geometry impractical for the rates required by high-speed flight.

**Figure 3 sensors-24-02204-f003:**
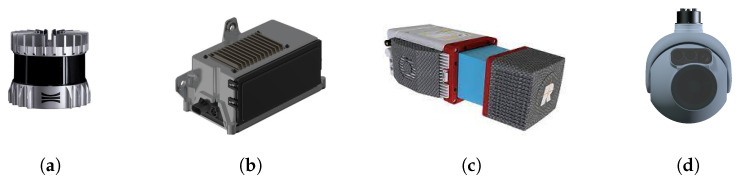
SeveralLiDAR sensors were evaluated for inclusion on the platform. Left to right are shown (**a**) the Ouster OS1, (**b**) the HRL131, (**c**) the RIEGL miniVUX-HA, and (**d**) the L3 Harris Tactical Geiger-Mode LiDAR sensors.

The data stream, resulting from the combination of the measurement method and measurement speed, requires highly accurate flight pose tracking over long distances at high speeds for the accurate integration of data into a unified 3D map. Achieving this may pose challenges considering the tracking accuracy limitations of onboard instruments, and substantial computation may be required for per-point or per-scan line motion compensation. Due to these reasons, the utilization of LiDAR sensing instrumentation for high-speed UAV mapping is not advisable.

#### 3.1.2. Event Cameras

Figure 4 shows several event camera sensors evaluated for inclusion in the platform payload. The key attractive aspect of event cameras that has sparked considerable interest from researchers and industry alike is the extremely high temporal accuracy. Specifically, event cameras can resolve intensity changes in the perceptual field at a temporal resolution of approximately 1 μs. For this reason, event cameras have been used in high-speed contexts.

While the deployment of event cameras as a component of UAVs may seem attractive due to the temporal resolution, there are several shortcomings associated with integrating this hardware into the UAV payload:Resolution: Resolution is a key parameter for depth accuracy as discussed in the stereo reconstruction Section 2.1.2. Accuracy strongly ties to both resolution and pixel size, δ, as shown in Figure 1b and Figure 2. Event camera resolution, 0.3 pixels, is a factor of 5–10 times lower than conventional image sensors, and the pixel size of δ = 18 μm is a factor of 6–18 times larger than conventional image sensors, e.g., the Sony IMX472 sensor has a resolution of 21 megapixels and a pixel size of 3.3 μm.Weight: While these sensors are lighter than LiDAR sensors, they weigh ∼100 g. and much lighter camera sensors are available.Latency: While the temporal resolution of event cameras is an impressive 1 μs, the latency of the measurements is on the order of <1 ms. This latency is similar to that of high frame rate conventional image sensors with frame rates of +100 fps and similar <1 ms latency.Nighttime Performance: Event cameras operate on similar principles to conventional visible light cameras. As such, they are suited to deployment in daytime contexts. The lack of an infra-red event camera requires completely separate perceptual software stacks for the vehicle in daytime and nighttime contexts.

Event cameras are a recent technology that has emerged and matured over the past decade. These sensors have unparalleled temporal resolution of 1 μs which makes them popular for capturing high-speed phenomena endemic to high vehicle speed applications. Yet, current technology has not matured to the extent required to make this sensor a viable option. Further, the development of a nighttime IR sensing event camera is an active area of sensor development under initiatives with no commercially viable examples. The drawbacks of having low sensor resolution, large pixel size, and no nighttime performance combined with comparable latency and weight to standard conventional cameras suggest that this sensor is not appropriate for inclusion as a component of the UAV payload for high-speed mapping applications.

#### 3.1.3. Electro-Optical and Infrared Cameras

Conventional image sensors, including electro-optical (EO) and infrared (IR) sensors, have many beneficial attributes that often make them the sensor of choice for perception designs that must satisfy low Size, Weight, and Power (SWaP) requirements. They are well suited for UAV mapping tasks for several reasons:SWaP: Both EO and IR camera modules are available commercially in a very large variety of form factors. This includes a compact 25 mm^3^ weighing 10–50 g requiring ∼1 W for power and providing temporally synchronized high framerate (60 fps) images.High-Quality Imaging: Modern cameras offer high-resolution imaging with the ability to capture fine details, which is crucial for mapping tasks, especially in scenarios where identifying objects is essential.Mapping and Geospatial Data: Cameras can be used for aerial imaging and photogrammetry to create detailed maps and 3D models of areas, making them valuable for urban planning, environmental monitoring, and disaster management.Stereo Vision: Cameras can be paired to create a stereo vision system. By capturing images from two slightly offset viewpoints, they can calculate depth information through triangulation, using the disparity between corresponding points in the two images. This method provides accurate 3D information.Integration with Other Technologies: Cameras can be integrated with other sensors and technologies, such as Inertial Measurement Units (IMUs) and Global Navigation Satellite System (GNSS), to enhance their capabilities and improve accuracy.Wide Field of View: Many cameras have wide-angle lenses or the ability to pan, tilt, and zoom (PTZ), providing a broad field of view and the flexibility to focus on specific areas of interest.Daytime and Nighttime Versatility: Camera sensors are capable of sensing in both daylight and nighttime conditions. If high sensitivity is needed in both scenarios, it is possible to replace daytime image sensors with infrared image sensors during nighttime conditions. This can be achieved with minor modifications to the underlying software and algorithms.Large Active Algorithm Ecosystem: Researchers worldwide develop cutting-edge algorithms for these sensors at top institutions. Utilizing this sensor type enables leveraging the latest, optimized, and theoretically advanced algorithms for vehicle perception tasks.Cost-Effectiveness: Compared to some other sensing technologies, cameras are cost-effective, making them accessible for a wide range of surveillance and mapping applications.

Conventional image sensors have many beneficial attributes that make these sensors attractive for multirotor UAV applications. These sensors have low SWaP requirements and can record >16 M measurements at a time from the environment. These sensors can be combined with lens components that provide both wide-angle viewpoints, e.g., a 230° FOV via the fisheye lens, for omnidirectional perception and confined viewpoints, e.g., 80° FOV “standard” lens, for high fidelity target tracking and mapping. The intensive work required to integrate these sensors and develop optimized algorithms to process their data to work using onboard computing resources can be reused between daytime (EO) and nighttime (IR) sensing contexts.

Image sensors designed for both infrared (IR) and visible light often share common image processing algorithms, including basic processes like filtering, noise reduction, contrast enhancement, and image registration. Additionally, object detection, recognition, feature extraction, and image fusion algorithms can typically be adapted for both IR and visible light images, leveraging shared features and patterns. However, notable differences emerge, primarily related to spectral characteristics, illumination, noise, calibration, temperature considerations, environmental conditions, and the unique sensitivities of IR images to object materials. These distinctions necessitate adjustments in algorithms to address variations in contrast, object recognition, and material discrimination, showcasing the need for specialized approaches in certain contexts.

#### 3.1.4. Recommendations

The assessment of available sensors suggests that the conventional image sensors are the best-practice sensors for multirotor UAV applications. EO and IR sensors, being lighter in weight and faster in measurement speeds compared to LiDAR sensors, also offer better image quality and nighttime measurement capability in contrast to event cameras. The choice of conventional sensors not only aligns with budgetary constraints but also caters to the diverse needs of UAV operations, encompassing navigation, mapping, and surveillance with exceptional performance and reliability.

### 3.2. Benchmark Dataset

A virtual environment that mimics real-world scenes was used for evaluation. Compared to real datasets, synthetic datasets for evaluating mapping algorithms bring a notable advantage in the form of readily available ground truth 3D models. This availability of ground truth data facilitates a more rigorous assessment of mapping performance, ensuring precise comparisons between the algorithm’s outputs and the known true state of the environment.

#### 3.2.1. Environment Simulation

AirSim [49] was used to simulate the dynamics of the drone. AirSim, developed by Microsoft, stands as a groundbreaking and influential simulator that has become a cornerstone in the development of autonomous drones and robotics. What sets AirSim apart is its capacity to simulate complex and dynamic environments with exceptional fidelity, replicating not only the physics of flight but also the intricacies of various sensors like cameras (RGB and depth), LiDAR, and GPS. Figure 5 shows an example of the AirSim simulated images captured by the cameras mounted on the multirotor.

The proposed approach used the Cesium plugin for the Unreal Engine, also known as “Unreal Cesium”, to simulate real-world scenes, and enhance the effectiveness of simulations. Although AirSim provides rich virtual environments for testing and fine-tuning a wide array of autonomous systems, these environments are often designed for games and lack realism. To synthesize virtual models that replicate real-world contexts, AirSim can be integrated into the Unreal Engine to allow the Unreal Cesium plugin to create digital twins of real-world environment models. The Cesium plugin, given the latitude and longitude coordinates of desired locations, can load 3D tilesets at the location from Google Maps in the AirSim simulator.

The Unreal Cesium plugin creates a powerful combination by integrating the Unreal Engine’s advanced rendering and simulation capabilities with Cesium’s geo-spatial visualization and data streaming features. By streaming high-resolution 3D models from Google Maps, overlying them onto real-world maps, and applying dynamic lighting and shadows, to provide precise representations of real-world locations, such as cities, terrains, and 3D models of buildings. This integration allows developers to create highly realistic and spatially accurate virtual environments for various applications. Once the 3D map is generated, it behaves as a collision object in the Unreal Engine. The UAV then interacts with this model using the geometry of the environment and a physics engine. Figure 6 demonstrates the benefits of the Unreal Cesium plugin for simulation. It shows three real-world locations: (1) the UNC Charlotte campus, USA, (2) the Grand Canyon, USA, and (3) Paris, France.

#### 3.2.2. Flight Simulation

Figure 7 depicts the proposed pipeline for flight simulation. QGroundControl (v4.3.0) and PX4-Autopilot (https://github.com/PX4/PX4-Autopilot/tree/98d893503495f7c28856bccf830082451b20265d accessed on 14 March 2024) are software components commonly used in the field of UAVs and drones. They work together to provide a comprehensive solution for controlling and managing drone flights. The missions are planned by defining waypoints, flight paths, and specific actions for the drone to perform and QGroundControl sends the mission plans to the autopilot system PX4-Autopilot. As an open-source flight control software for UAVs, PX4-Autopilot runs on the flight controller onboard the drone and is responsible for stabilizing the aircraft, executing flight plans, and interfacing with sensors and actuators. Controlled by PX4-Autopilot, the simulated UAV in AirSim follows the planned trajectory in a high-realism virtual environment created by Unreal Engine and Cesium plug-in. The cameras mounted on the UAV then capture the images (RGB, depth, etc) of the scene. These images, along with the ground truth vehicle odometry, can be obtained from AirSim, which then can be applied together to generate the ground truth 3D model of the world.

#### 3.2.3. Ground Truth Geometry

The ground truth geometry is generated by applying AirSim’s built-in functionality for ground truth pose and noiseless telemetry to collect color-attributed point cloud data from simulated noiseless pixel-aligned RGB and depth images captured by a drone that traverses the environment. The telemetry is extracted from Cesium Tile data for generating high-fidelity geometric models. Poses and point clouds are integrated using standard mapping methods to reconstruct the scene geometry. Figure 8 shows an example of integrating the drone odometry (shown as the red curve Figure 8c) and the point cloud from RGB-D image sequences to create a geometric model of the scene.

### 3.3. Evaluation Methods

Mapping algorithm performance is evaluated using the following two key performance criteria: (1) mapping accuracy, and (2) mapping speed. Mapping accuracy is assessed by comparing the geometry of the reconstructed point cloud with the ground truth point cloud. Geometric accuracy measures each mapping algorithm’s ability to faithfully capture spatial relationships in the environment. Algorithm performance speed quantifies the amount of 3D estimates generated per unit of allocated computational resources. More computation allows more points to be tracked in sequential frames and the creation of more keyframes. Both tracked points and keyframe data feed non-linear bundle adjustment and batch trajectory optimization processes which improve map fidelity but can require significant computational resources. The evaluation methods allow result analysis that indicates design trade-offs associated with each mapping algorithm.

#### 3.3.1. Point Cloud Registration

Point cloud registration seeks to compute the alignment between two 3D point clouds measured from the same surfaces in distinct coordinate systems. Alignment algorithms identify point correspondences between the misaligned point cloud datasets and compute the rigid Euclidean transformation that makes corresponding points coincide. To accomplish this, a point cloud is selected as the static dataset and all other measured point clouds are transformed to align with the measurement coordinate system of the staticdataset. Figure 9 shows an example of registering two point clouds where the red point cloud is the source set and the blue is the static dataset.

The Iterative Closest Point (ICP) algorithm [50] is employed to estimate point cloud alignments. The ICP algorithm consists of the following two steps: (1) compute correspondences and (2) compute the best alignment given the correspondence. Steps (1) and (2) are iterated until the alignment of Step (2) stops changing. Correspondences for a given iteration are calculated by finding the closest point in the static dataset to each point in the dataset being aligned. Closest point searches stop at a user-specified search radius for each point. The ICP algorithm seeks to minimize the RMSE (Root Mean Square Error) of all the distances between corresponding points and terminates when the gradient of RMSE is below a predefined threshold or a predetermined maximum iteration count is reached. Alignments resulting from the ICP algorithm are used to evaluate the geometry accuracy of mapping algorithms.

**Figure 9 sensors-24-02204-f009:**
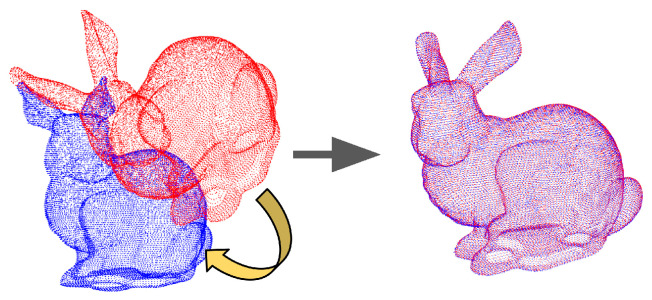
An example of point cloud registration [51]. Red: source point cloud. Blue: target point cloud. Purple: registration result.

#### 3.3.2. Geometric Accuracy

With the correspondences found using the ICP algorithm, the geometric accuracy of the reconstructed map is measured by the distance between corresponding points in the ground truth 3D model and the reconstructed 3D map. The mean of this distance of all the corresponding points is used to evaluate the accuracy performance, calculated as follows:(5)$$\bar{x}=\frac{1}{N}\sum_{i=1}^{N}\left|\right|P_{i}-T_{i}\left|\right|$$
where *N* is the total number of corresponding points, $P_{i}$ is the position of the *i*-th corresponding point, and $T_{i}$ is the ground truth (reference) position for that point.

Further, the standard deviation (std) of the errors (distances) is used to evaluate the variability in the errors, calculated as follows:(6)$$\sigma =\sqrt{\frac{\sum_{i=1}^{N}\left(|\right|P_{i}-T_{i}\left|\right|-\bar{x}{)}^{2}}{N}}$$

#### 3.3.3. Computational Cost

To assess the computational efficiency of mapping algorithms, two key metrics were focused on in this article: keyframe creation time and frame tracking time. These metrics were selected to provide insights into the mapping speed of the algorithms.

Keyframe Creation Time: Keyframe creation time quantifies the time required to identify keyframes during the mapping process. Keyframe creation is arguably the most time-consuming process of the mapping pipeline, often 5–10× slower than tracking [15]. Creating too many keyframes will cause the system to eventually lag behind the frame rate. Keyframe creation time reflects the computational efficiency of map reconstruction.

Frame Tracking Time: Frame tracking time represents the duration required for the algorithms to process and track individual frames with respect to the keyframes. This metric reflects the algorithm’s ability to track and update the mapping information in real-time.

These two metrics collectively provide a comprehensive evaluation of the computational cost of mapping algorithms. The results of these evaluations are discussed in Section 4.2, providing the relative efficiency and performance trade-offs among the implemented algorithms.

## 4. Results

The experimental scene was a virtual model of the UNC Charlotte campus near the football stadium. The model was generated using AirSim and the underlying Unreal Engine in combination with the Cesium Tiles plugin.

Experiments were conducted on an Intel i7-12700KF CPU. The implementations of DSO and DSOL that were made available on GitHub by the authors were used [52,53]. SDSO implementation by the authors is not available so an open-source third-party implementation on GitHub was chosen [54]. All implementations adhered to the original configuration optimized by their authors for accuracy and/or speed performance including the number of active keyframes and maximal tracking points per frame. Customized modifications made to all three algorithms respectively for collecting experimental data include the following: (1) saving the generated point cloud to a PCD file; (2) saving the keyframe ID and associated creation time to a text file; (3) saving the frame ID and associated tracking time to a text file.

Experiments consist of a simulated quadrotor vehicle that traverses the virtual scene at heights ranging from 12 m to 20 m and at speeds ranging from 16.5 m/s to 20 m/s. During the flight camera sensor telemetry was recorded from a stereo pair of camera mounts to the UAV chassis. Algorithms processed the telemetry to generate mapping data for the environment. The SfM algorithm (DSO) used data from the left camera of the stereo rig while DSOL and SDSO (stereo reconstruction) utilized all the available image data. The left camera is chosen to define the sensor coordinate system and the noiseless depth sensor is co-located with the left camera to record ground truth depth for each pixel measured within the view of the left camera. Figure 10a illustrates the simulated drone’s flight over the UNC Charlotte football stadium, covering a 3-min flight duration and capturing 1027 RGB stereo pair frames and associate ground truth depth. Sample images from the drone’s simulated RGB and depth sensors are displayed in Figure 10b,c.

### 4.1. Mapping Accuracy Evaluation

Using the methods of Section 3.2.3, a ground truth point cloud of the experimental scene was calculated which is shown in Figure 11. This point cloud serves as the ground truth geometry to evaluate the mapping accuracy of different algorithms. Figure 12 shows the point cloud respectively reconstructed by DSO, SDSO, and DSOL. All three maps qualitatively encode the shape and size of objects from the experimental scene. However, the point density and scene details of DSO and DSOL maps outperform the DSOL map.

#### 4.1.1. Quantitative Analysis

Table 1 details the density and accuracy characteristics of mapping results obtained from different algorithms. The ICP algorithm was used to align estimate maps with the ground truth point cloud. Criteria for alignment convergence and correspondence calculation included the following: (1) a search radius of 0.5 m, (2) algorithm termination criteria which are triggered when either the RMSE of corresponding points changes by less than 0.00001 or the maximum number of iterations exceeds 1500. Alignment results allow for statistics to be tabulated on the point cloud accuracy for each algorithm. DSO, as a monocular SfM algorithm, is unable to estimate the scene scale accurately. The scale factor was estimated from the ICP algorithm for DSO and the DSO mapping result (point cloud) was scaled by the estimated factor for accuracy evaluation. For SDSO and DSOL, the scale factor was not estimated since the scale of the scene can be derived from the stereo data. In our experiments, the estimated map scale of DSO was 35.98. Table 1 captures key statistics of the alignment process for the three algorithms evaluated. Each row of this table is explained below:points: Total points in the reconstructed point cloud.correspondences: Total amount of correspondences.mean: Mean of the distance between all corresponding points.std: Standard deviation of the distance between all corresponding points.

**Table 1 sensors-24-02204-t001:** Quantitative evaluation of the point clouds generated by three mapping algorithms. DSO and SDSO maps contain more points and correspondences than the DSOL map. The error statistics (“mean” and “std”) indicate both higher accuracy and consistency in the DSO and SDSO mapping results than DSOL. The DSO map was scaled by a factor (35.98) estimated from the ICP algorithm.

	DSO	SDSO	DSOL
points	204,345	212,179	6662
correspondences	172,862	183,460	2799
mean (m)	0.110	0.110	0.177
std (m)	0.110	0.111	0.145

Notably, DSO and SDSO maps, similar to each other in the total amount of points, encompass ∼30 times more points than the DSOL map. This aligns with the observations in Figure 12. The difference in the correspondence sets is more pronounced, with DSO and SDSO revealing ∼65 times more correspondences than DSOL. DSO, after scaling, follows very closely to SDSO in terms of mapping accuracy performance, while both of them exhibit a 60.9% lower mean error than DSOL and a 31.8% smaller standard deviation. DSO and SDSO prove more accurate and consistent in their mapping results, while DSOL lags in terms of both precision and reliability.

Figure 13 depicts the distribution of the distance between corresponding map locations for the three algorithms. DSO and SDSO exhibit similar distributions and most 3D measurements lie within 0.15 m to their corresponding location in the ground truth model. In contrast, DSOL has significantly fewer points within the 0.15 m distance range and a nearly constant number of points having similar errors for greater distances. This supports the mapping accuracy results shown in Table 1.

Figure 14 indicates the capability of each algorithm to estimate large depths from a given viewpoint (Figure 14a) and expected error for a depth estimate for each depth (Figure 14b) where depths have been binned to 5 mintervals for tabulation.

Figure 14a shows the distribution of depth values for the keyframes of the trajectory which are responsible for generating depth values. Figure 14a indicates that a majority of depth estimates range from 20 m to 60 m. One can also see that DSO is capable of generating estimates at larger depths than the two other algorithms (see ranges 100–130 m). DSOL tends to reconstruct points within 60 m and shows a slightly bimodal behavior with a high population of measurements in the 60–100 m range which may be an artifact due to the experimental context. Figure 14b portrays the expected depth error in each keyframe. Figure 14a also lacks any presence of short ranges. This can be attributed to flying at low-altitude where most data is further than 10 m away.

Figure 14b, shows the expected depth error for estimate depths. Inspection of the results for distances of 20–40 m, the reconstruction error of DSOL is approximately 0.15 m per point while DSO and SDSO are close to each other having an error of approximately 0.085 m. DSO outperforms SDSO across most depth ranges with slightly smaller distance measurements. Additionally, the error distributions exhibit a quadratic growth pattern as predicted by theoretical models as described in Figure 2. High error is noted at short ranges of less than 25 m. This can be attributed to a lack of sufficient supporting image data due to the high velocity of the UAV. Surfaces close to the vehicle move quickly through the field of view and exhibit more motion artifacts leading to higher depth estimation error.

Figure 14 indicates that DSO exhibits lower error values across all ranges yet has fewer points. This can be attributed to a strong filter on the acceptable point depth covariance for map points within the algorithm. SDSO and DSOL exhibit lower accuracy compared to those produced by DSO.

#### 4.1.2. Qualitative Analysis

Figure 15 shows reconstructed maps from DSO, SDSO, and DSOL. A qualitative examination of these results unveils notable distinctions in their alignment with the ground truth. DSO and SDSO, with their significantly higher point densities appear to exhibit good accuracy as evidenced by the details of the road network that have been captured and include intricate and well-aligned geometries, e.g., road curbs. The enhanced point density, particularly evident in the football stadium region, allows for a more detailed reconstruction and appears to provide better alignment results relative to ground truth here. Conversely, DSOL, characterized by a sparser point cloud provides a reduced level of detail, particularly in complex structures like the football stadium. Although DSOL shows good alignment for roads, the sparsity of the estimate limits the map details.

### 4.2. Computational Cost Evaluation

Computation cost for the considered algorithms considers the resources required by two critical mapping algorithm functions cost: (1) keyframe creation time and (2) frame tracking time. These metrics serve as crucial benchmarks in assessing the algorithms’ ability to swiftly and accurately generate keyframes, as well as tracking real-time camera pose changes during the mapping process. Through this examination, we seek to offer valuable insights that contribute to the informed selection and deployment of mapping solutions for low-altitude UAV flights, particularly for high-speed applications.

#### 4.2.1. Keyframe Creation Time

Figure 16 illustrates the keyframe creation time for DSO, SDSO, and DSOL. It can be seen that SDSO requires the most time to create a keyframe, averaging ∼220.73 ms per keyframe, as reported in Table 2. DSO incurs lower computational cost for keyframe creation since the stereo disparity map estimation algorithm is not required resulting in keyframe times averaging around ∼200.28 ms per keyframe. DSOL requires ∼7.39 ms per keyframe which is approximately 30 times faster than competing approaches. This can be attributed to the simplified keyframe creation process facilitated as a combination of a simplified disparity computation algorithm and parallel processing. The columns in Table 2 delineate the statistical distribution of keyframe creation times, including the minimum, maximum, and mean values, with the “std” column denoting the standard deviation. In summary, SDSO necessitates 10.21% more keyframe creation time than DSO and 2886.87% more than DSOL, while DSO requires 2610.15% more time than DSOL.

Table 2 contains data that provides quantitative measures for the aggregate number of keyframes generated by the three algorithms (“total kfs” column). DSOL generates approximately ∼80% fewer keyframes compared to its counterparts which can be attributed to slightly more restrictive requirements for keyframe creation. Noteworthy is the observation that DSO creates 29 fewer keyframes than SDSO. This discrepancy is attributed to a delayed initialization of the DSO system, commencing at the 50th frame in our experiments, in contrast to the immediate initialization of SDSO and DSOL. Such delay is also illustrated in Figure 16 as the DSO curve starts later than SDSO and DSOL. The DSO initialization process relies on assigning random depth values to candidate points and predicting the initial camera movement pattern, demanding precise assumptions about initial depth values and camera motion. In contrast, mapping systems employing stereo cameras, such as SDSO and DSOL, leverage stereo matching for enhanced depth initialization, leading to increased accuracy. Divergence in keyframe quantities among the algorithms also mirrors the disparities in point cloud density depicted in Figure 12, given that these points are derived from the keyframes.

#### 4.2.2. Frame Tracking Time

Figure 17 depicts the frame tracking time across various algorithms, employing scatter points for visualization. Results show the very high performance achieved by DSOL which requires very little computation for each tracked frame. In the case of DSO and SDSO, the tracking time is stratified into two distinct regions. The upper region, requiring approximately ∼60 ms for tracking, corresponds to keyframes, while the lower region, with an average tracking time of ∼20 ms per frame, pertains to non-keyframes. This 3× difference in tracking time arises from the creation of a new keyframe, where existing point tracks must be terminated and a collection of new point tracks must be initialized incurring significant computational cost to transfer the tracking information. Subsequent frames are then exclusively tracked to this keyframe, employing traditional two-frame direct image alignment methods. This stratification in tracking time offers insights into the computational demands associated with keyframe and non-keyframe tracking, highlighting the intricacies involved in SfM methods that must maintain accurate and efficient tracking across consecutive frames.

Figure 17 also shows two apparent bands in the lower region for results of DSO and SDSO. The higher band characterizes the tracking time for frames immediately succeeding keyframes, while the lower band denotes the tracking time for other frames. A repeated pattern exists where ∼5 ms of addition time is required to process frames following keyframes. Figure 17b zooms into a subsection of the data associated with frame indices 290–310. Close examination of this phenomenon indicates that newly formed tracks require more time as the points of the initial keyframe have to be sorted into reliable and unreliable tracks thereby necessitating slightly more computation.

## 5. Conclusions

This paper presents a study on low-altitude and high-speed drone applications. An examination of various sensors underscored their strengths and challenges, guiding the selection of suitable devices for specific operational scenarios. The experiments centered on evaluating three prominent mapping algorithms—DSO, SDSO, and DSOL—in a simulated environment, providing valuable insights into the performance of these mapping algorithms. Each algorithm exhibits unique strengths and trade-offs, catering to specific requirements in UAV-based mapping scenarios. DSO, operating as a monocular mapping algorithm, demonstrates versatility in capturing scenes with a single camera, albeit with limitations in scale estimation. SDSO, incorporating stereo depth perception, excels in accuracy and spatial fidelity, as evidenced by its superior point cloud density and detailed reconstructions, particularly in complex structures like the football stadium. On the other hand, DSOL, designed for efficiency, streamlines the mapping process, offering reliable reconstructions with reduced computational demands. The findings suggest that, in cases where UAVs have limited computing resources, DSOL emerges as the optimal choice. For systems equipped with payload capacity and moderate compute resources, SDSO proves to be the most suitable option. When dealing with a single camera, DSO is the preferred choice for applications demanding dense mapping results.

Future work may involve refining these algorithms for optimized performance in diverse environments, ultimately contributing to advancements in UAV-based mapping for low-altitude and high-speed drone applications. This study contributes to the ongoing discourse on mapping algorithms, providing valuable insights for researchers and practitioners navigating the dynamic landscape of UAV applications in remote sensing and environmental monitoring.

## Figures and Tables

**Figure 1 sensors-24-02204-f001:**
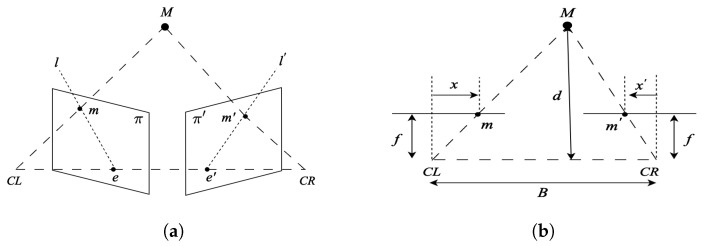
(**a**) Epipolar geometry of two cameras. (**b**) Epipolar geometry of a rectified image pair.

**Figure 2 sensors-24-02204-f002:**
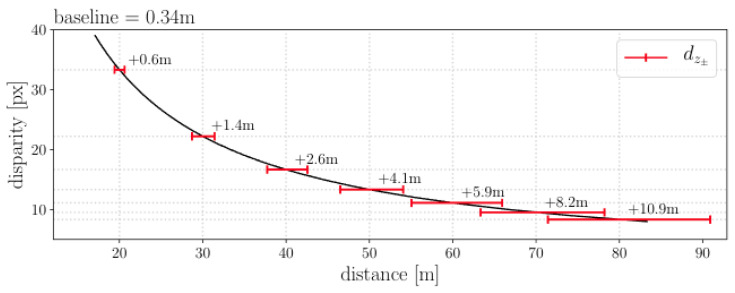
The dependency between depth estimation accuracy and the baseline of the stereo camera design for a baseline, *B* of 34 cm, based on [34].

**Figure 4 sensors-24-02204-f004:**
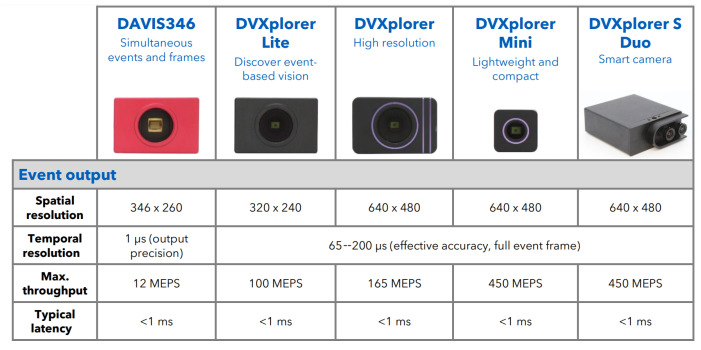
A collection of event cameras commercially available from the iniVation Corp [48].

**Figure 5 sensors-24-02204-f005:**
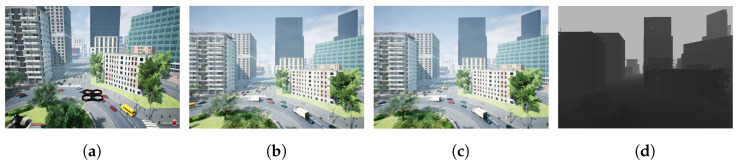
An example of an AirSim city environment showing the following: (**a**) the FPV view in the simulator where the drone is hovering, (**b**) RGB image from the simulated left camera mounted on the drone, (**c**) RGB image from the simulated right camera, and (**d**) depth image from the simulated depth sensor where objects closer to the depth camera appear darker. (**a**) AirSim simulator; (**b**) left camera image; (**c**) right camera image; (**d**) Depth image.

**Figure 6 sensors-24-02204-f006:**
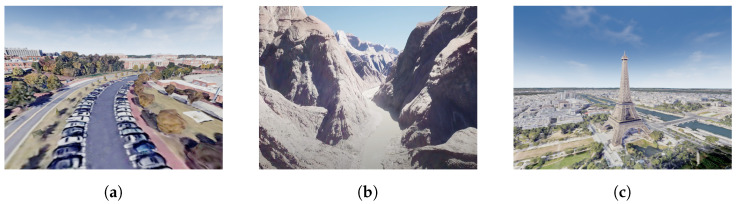
Realistic environments created using Unreal Engine and Cesium Plug-in. (**a**) UNC Charlotte, NC, USA; (**b**) Grand Canyon, AZ, USA; (**c**) Eiffel Tower, Paris, France.

**Figure 7 sensors-24-02204-f007:**
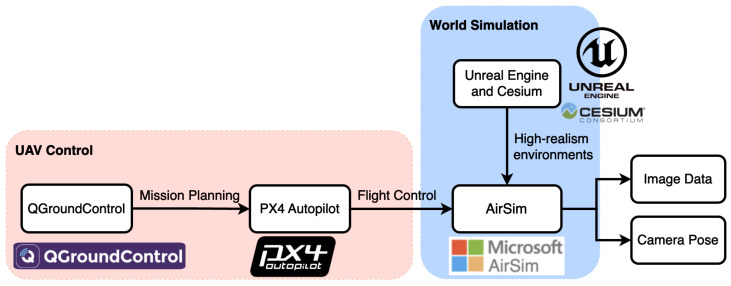
The flight simulation pipeline integrates the following four robot development technologies to facilitate development and testing: (1) Unreal Engine and Cesium plugin (high-realism image synthesis), (2) AirSim (vehicle dynamics), (3) QGroundControl (mission planning), and (4) PX4-Autopilot (vehicle control and Software-In-The-Loop).

**Figure 8 sensors-24-02204-f008:**
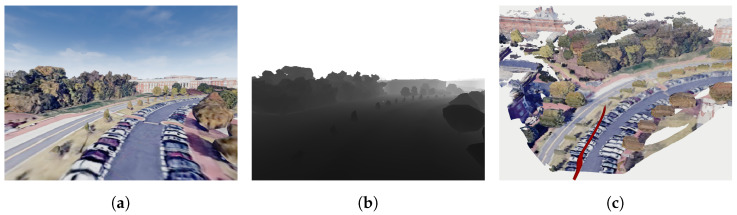
Ground truth geometry can be generated by transforming the point clouds of RGB-D frame sequences to the odometry of the drone which is shown in red in Figure 8c. (**a**) An RGB frame captured by the drone; (**b**) A depth frame captured by the drone; (**c**) Integrating pose and point clouds to generate a map.

**Figure 10 sensors-24-02204-f010:**
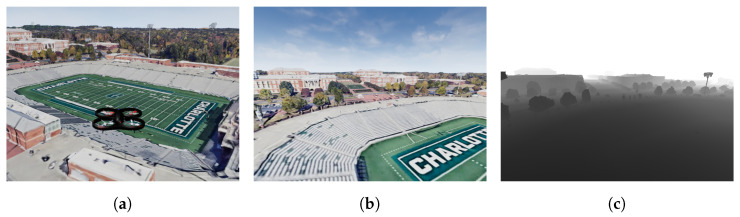
A simulated UNC Charlotte campus world. (**a**) A quadrotor flying in a virtual model of UNC Charlotte; (**b**) A RGB image captured by the drone camera; (**c**) A depth image captured by the drone camera.

**Figure 11 sensors-24-02204-f011:**
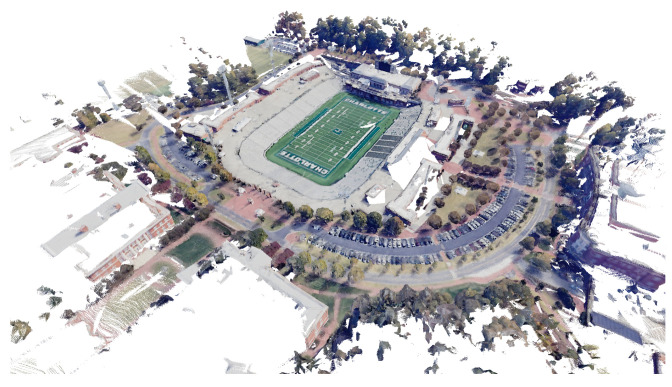
The ground truth point cloud of the scene generated by applying the ground truth odometry to the point cloud of each RGB-D frame.

**Figure 12 sensors-24-02204-f012:**
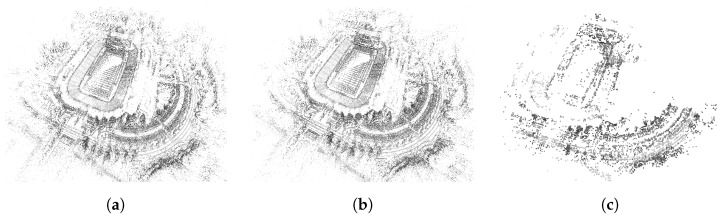
Point clouds generated by (**a**) DSO, (**b**) SDSO, and (**c**) DSOL. DSO and SDSO generated much more point clouds than DSOL. The color of the points is represented by the grayscale color of the scene point.

**Figure 13 sensors-24-02204-f013:**
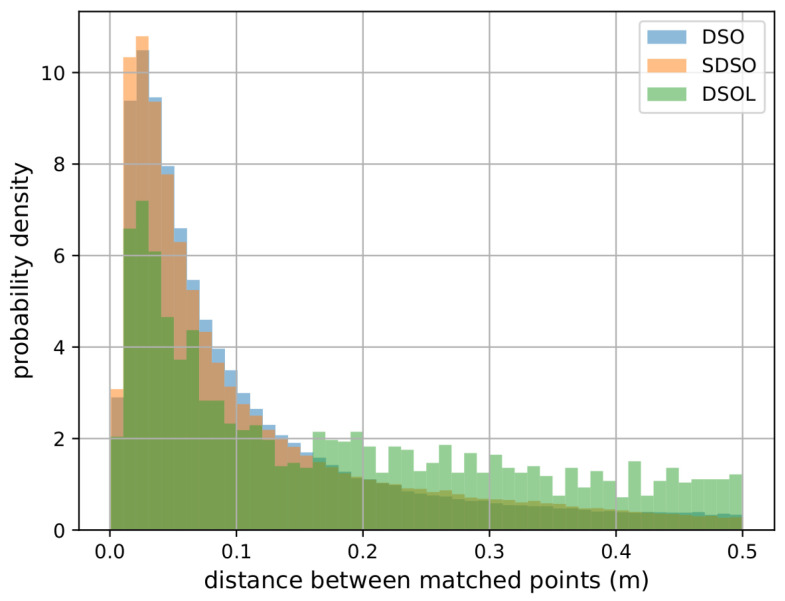
Quantitativeanalysis of the distribution of the closest point distances for correspondences from the mapping results to the ground truth point cloud. DSO and SDSO have similar distributions while the mean correspondence error is larger in DSOL.

**Figure 14 sensors-24-02204-f014:**
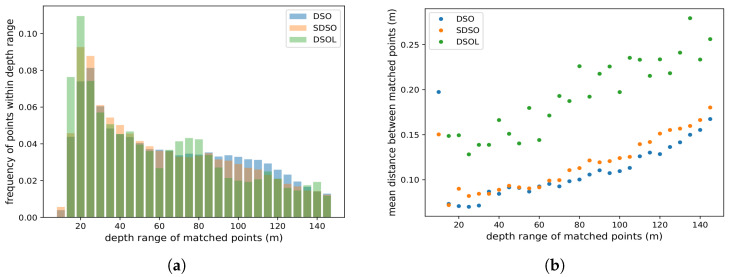
(**a**) Distribution of point depth in the reconstructed maps and (**b**) the average distance between the matched points at different depth ranges.

**Figure 15 sensors-24-02204-f015:**
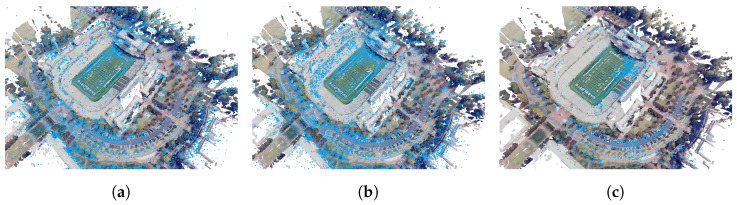
Reconstructed point clouds (blue) overlaid with the ground truth point cloud (actual color): (**a**) DSO, (**b**) SDSO, and (**c**) DSOL. DSO point cloud has been scaled by the factor estimated by ICP.

**Figure 16 sensors-24-02204-f016:**
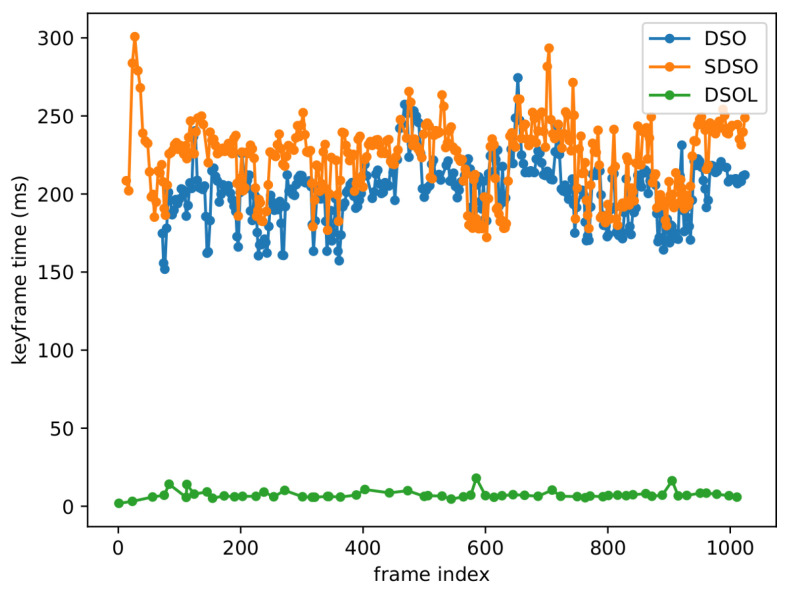
The points plotted along the curves represent the keyframe creation time at each frame. Stereo DSO (SDSO) exhibits a higher temporal requirement than monocular DSO, demonstrating significantly greater computational overhead than DSO-Lite (DSOL). The density of points on the curves serves as a visual indicator of the number of keyframes generated, revealing that both DSO and SDSO produce a larger quantity of keyframes than DSOL.

**Figure 17 sensors-24-02204-f017:**
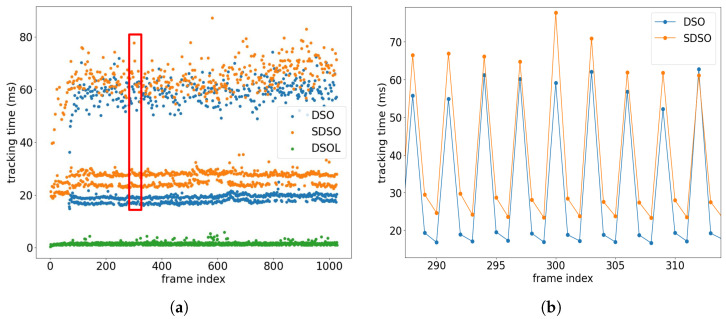
Frame tracking time of different algorithms. (**a**) shows the tracking time for all frames including keyframes and non-keyframes. The data are plotted as scatter points for a clear visualization. (**b**) shows the DSO and SDSO tracking time for frames 290∼310 which corresponds to the region highlighted by the red box in (**a**).

**Table 2 sensors-24-02204-t002:** Statistics of the keyframe creation results of three mapping algorithms. The “total kfs” column shows the total amount of the keyframes created by three algorithms. The “min”, “max”, and “mean”, respectively, show the minimum, maximum, and average time for keyframe creation. The “std” column denotes the standard deviation of the keyframe creation time.

	Total kfs	min (ms)	max (ms)	mean (ms)	std (ms)
DSO	330	151.83	274.44	200.28	19.63
SDSO	359	172.19	300.79	220.73	23.08
DSOL	61	1.88	17.99	7.39	2.66

## Data Availability

The raw data supporting the conclusions of this article will be made available by the corresponding author upon request.

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
