# Peer review of "UAV-Borne Mapping Algorithms for Low-Altitude and High-Speed Drone Applications"

_sensors, 2024, doi:10.3390/s24072204_

Round 1

Reviewer 1 Report

Comments and Suggestions for Authors     Some of the references are old, such as ones that are from years 1991, 2004, 2007, 2008, and 1992. The authors are encouraged to use more recent references.    Fig. 4, 5, 7, and 17 are not clear   Fig. 5 and 17 should be split into multiple  figures or images   Fig.3 should include letter captions for images     The authors should compare their findings and results with other previous related works  Comments on the Quality of English Language

proofreading needed

Reviewer 2 Report

Comments and Suggestions for Authors

This article analyzes current state-of-the-art sensors and how these sensors work with several mapping algorithms for UAV (Unmanned Aerial Vehicle) applications, focusing on canopy-level and high-speed scenarios. A new experimental construct is created using highly realistic environments, which is made possible by integrating the AirSim simulator with Google 3D maps models using the Cesium Tiles plugin. Experiments are conducted in this high-realism simulated environment to evaluate the performance of three distinct mapping algorithms: 1. Direct Sparse Odometry (DSO), 2. Stereo DSO (SDSO), and 3. DSO Lite (DSOL). Experimental results evaluate algorithms based on their measured geometric accuracy and computational speed. The results provide valuable insights into the strengths and limitations of each algorithm. Findings quantify compromises in UAV algorithm selection, allowing researchers to find the mapping solution best suited to their application, which often requires a compromise between computational performance and the density and accuracy of geometric map estimates. 

Reviewer 3 Report

Comments and Suggestions for Authors

The manuscript is well-written and easy to follow. However, there are still several issues to be addressed:

(1) The author mixed the definitions of SLAM and SfM. If we say SfM, it is usually referring to a post-processing. So please revise the definition. DSO, SDSO, DSOL are all real-time algorithms.

(2) The author highlights the canopy level, please simulate the environment in the forest to demonstrate the performance of the different methods.

(3) In the UAV LiDAR section, please review more related works:

Li, J., Yang, B., Chen, C., & Habib, A. (2019). NRLI-UAV: Non-rigid registration of sequential raw laser scans and images for low-cost UAV LiDAR point cloud quality improvement. ISPRS Journal of Photogrammetry and Remote Sensing158, 123-145.

Nguyen, T. M., Cao, M., Yuan, S., Lyu, Y., Nguyen, T. H., & Xie, L. (2021). Viral-fusion: A visual-inertial-ranging-lidar sensor fusion approach. IEEE Transactions on Robotics38(2), 958-977.

Round 2

Reviewer 3 Report

Comments and Suggestions for Authors

The author has answered my questions. Now it could be accepted.